# The American Society of Anesthesiologists Physical Status (ASA-PS) Risk Group Classification Can Be Used to Anticipate Functional Recovery Outcomes after the Surgical Treatment of Female Urinary Incontinence with Transobturator Suburethral Tape

**DOI:** 10.3390/jcm9082607

**Published:** 2020-08-11

**Authors:** Juan Cisneros-Pérez, Gemma Yusta-Martín, María-Pilar Sánchez-Conde, Barbara-Yolanda Padilla-Fernandez, Lauro-Sebastian Valverde-Martínez, Mario Martin-Hernandez, Sara Prieto Nogal, Javier Flores-Fraile, Manuel Esteban-Fuertes, María-Begoña García-Cenador, María-Fernanda Lorenzo-Gómez

**Affiliations:** 1Department of Surgery, University of Salamanca, 37007 Salamanca, Spain; juancisneros@usal.es (J.C.-P.); gyusta70@hotmail.com (G.Y.-M.); pconde@usal.es (M.-P.S.-C.); sebasv_2000@hotmail.com (L.-S.V.-M.); mbgc@usal.es (M.-B.G.-C.); mflorenzogo@yahoo.es (M.-F.L.-G.); 2Department of Anesthesiology of University Hospital of Salamanca, 37007 Salamanca, Spain; 3Section of Urology, Department of Surgery, University of La Laguna, 38200 San Cristóbal de La Laguna, Spain; padillaf83@hotmail.com; 4Renal Urological Multidisciplinary Research Group (GRUMUR) of the Institute of Biomedical Research of Salamanca (IBSAL), 37007 Salamanca, Spain; mariomhdez@yahoo.es; 5Department of Urology, University Hospital of Ávila, 05071 Ávila, Spain; sarab.prieto@hotmail.com; 6Department of Urology of National University Hospital of Paraplegic, 45004 Toledo, Spain; estebote@telefonica.net; 7Department of Urology of University Hospital of Salamanca, 37007 Salamanca, Spain

**Keywords:** postoperative complications, stress urinary incontinence (SUI), suburethral sling, recovery of function

## Abstract

Introduction: Stress urinary incontinence (SUI) has an incidence of 15–80% in women. One of the most widely used surgical techniques for treatment is the placement of a suburethral transobturator tape (TOT). Although this technique has a relatively low morbidity rate, it is not exempt from intraoperative or postoperative complications, which can have an impact on functional recovery, understood as the return to routine life prior to the intervention. Aims: To assess the time for functional recovery in women operated on for SUI by TOT; to identify complications and related factors, according to anaesthetic risk, which condition the time to functional recovery; and proposals for improvements in the prevention of possible complications and in reducing functional recovery time. Materials and methods: A non-concurrent prospective observational multicenter study of 891 women undergoing TOT for stress urinary incontinence since 1 April 2003, who were successful in achieving urinary continence (completely dry). Study groups: GA (*n* = 443): patients with ASA I risk. GB (*n* = 306): patients with ASA II risk. GC (*n* = 142): patients with anaesthetic risk ASA III. Investigated variables: age, body mass index, follow-up time, secondary diagnoses, surgical history, obstetric-gynecological history, toxic habits, and complications derived from surgery: bleeding, pain, infection. Descriptive statistics, Student’s t test, Chi2, Fisher, ANOVA, multivariate analysis, significance for *p* < 0.05. Results: Mean age was 60.10 years (SD13.38), with no difference between groups. Mean body mass index (BMI) was 26.55 kg/m^2^ (SD 4.51), lowest in GA. GB had more HT (38.6%) than GC (23.23%), more type 2 diabetes (19.83% versus 10.56%), and more respiratory disorders (6.97% versus 2.11%). There were more women with anxiety in GB (19.3%) than in GC (6.33%) (*p* = 0.0221) and GA (10.51%) (*p* = 0.0004). There was more hypothyroidism in GB (16.08%) compared to GC (2.11%) and GA (9.07%). There was more history of curettage in GC (11.97%) versus GB (5.63%); and more pelvic surgery in GB (71.31%) and GC (66.9%) compared to GA (32.57%). There were more concomitant treatments with benzodiazepines in GC (27.46%) and GB (28.41%) than in GA (8.86%), and more parapharmacy treatments in GB (17.96%) than in GC (6.33%). Following the operation, 113 patients had some sign or symptom that required medical attention: in GA 48 (10.83%), in GB 49 (16.06%), in GC 16 (13.22%). Mean days until functional recovery in patients with complications: in GA 5.72 (SD2.05); bleeding 3 (SD1), pain 6.40 (SD1.34), and infection 7.33 (SD0.57), with fewer days for bleeding than for pain or infection. GB: 27.96 (SD 28.42), bleeding 3 (SD0), pain 46.69 (SD31.36), infection 10.83 (SD3.90); lowest for patients with bleeding. GC: 9.44 (SD 2.50); for bleeding 7.66 (SD2. 08), pain 10.66 (SD1.15), infection 10 (SD3.46); no differences. Overall, for women with bleeding, the time was 4.16 days (SD1.94); less in GA and GB than in GC. Pain, at 31.33 days (SD 30.70), was the factor that most delayed functional recovery; in GB women, it took longer to return to work due to pain (45.96, SD31.36) compared to GA (6.4, SD 1.34) and GC (10.66, SD1.15). In women with infection, overall mean time was 10.11 days (SD 3.61) with no difference between groups. Conclusions: Mean time for the return to normal activity in patients who underwent TOT for SUI is 5 days if there are no complications, and 16.91 days if there are any. The ASA-SP risk group classification can be used to anticipate functional outcomes. An ASA-PS risk-based functional recovery forecasting protocol should be adapted, especially ASA II patients who may present with long-term disabling postoperative pain. Preventive management measures are proposed that favour functional recovery.

## 1. Introduction

Stress urinary incontinence (SUI) is the involuntary loss of urine that occurs when urethral resistance is overcome by abdominal pressure [1]. It is a frequent pathology in women, with an incidence rate that fluctuates between 15 and 80% [2]. The prevalence and severity of SUI increases with age [3].

The options for treating SUI include various types of treatments and will depend on the age and comorbidities of the patient. Among these treatments are physical therapy, pessaries, urethral bulking injections and, more often, surgery. Among the most widely used surgical techniques is the implantation of tapes below the mid-urethra, such as those placed by the transobturator route (TOT), attributed with a low incidence of complications, a short operative time, a fast learning curve and a success rate of over 80% [4]. Worldwide, suburethral sling implants are used for treating SUI in an estimated 9.45 cases in every 10,000 women [5].

Although all types of suburethral slings are considered minimally invasive procedures, they are not without intra- or postoperative complications, which have a significant impact on the final evaluation of the effectiveness of the procedure [6]. The reported complication rate ranges from 3.5 to 5% [7,8,9].

In the decision-making process of surgeons and patients, the ability to predict postoperative complications has a limited influence [10]. The risk from the participation of resident physicians in surgery, or of the obesity and age of the patient, have been studied to quantify the risk of complications [11]. Risk calculators have been developed, but they are complex and include variables that are not available for all patients [12].

The American Society of Anesthesiologists Physical Status Classification System (ASA-PS) describes the general health of a patient [13]. ASA risk has been reported to be associated with postoperative morbidity and mortality [11,14], serves as a predictor of length of hospital stay, and has been correlated with the development of postoperative complications such as bleeding or infection of the surgical wound [13,15].

Studies that assess functional recovery after implantation of a transobturator suburethral tape are limited [4,16]. Additionally, different questionnaires are often used, making it difficult to compare the results.

Functional recovery is defined as the physical recovery and functional progression of all aspects affected by an injury, and is a fundamental element for returning to work or normal activity under the best possible conditions, safely and painlessly [17].

Due to the limited number of studies on functional recovery after SUI surgery by the TOT procedure, and the factors that influence recovery, we proposed this study with the aim of discovering the relationship between ASA-PS anaesthetic risk groups, postoperative complications and functional recovery after TOT intervention to correct SUI in women.

The objective of this study is to assess recovery time in patients treated for SUI via TOT. Identify the factors that influence functional recovery time after surgery and determine the most frequent postoperative complications. Establish the relationship between anaesthetic risk groups and functional recovery. Proposals for improvement for the prevention of possible complications and reducing functional recovery time in these patients.

## 2. Materials and Methods

### 2.1. Design

Multicentre, prospective non-concurrent, observational study.

### 2.2. Sampling

The sample selection consisted of 891 women who had undergone TOT by SUI since April 2003 with an exhaustive successive selection, and who achieved total urinary continence, on a total base sample of 1000 patients operated on by SUI. Patients who failed to achieve urinary continence were excluded to avoid bias.

Continence was evaluated postoperatively at months 1, 6, and 12, and annually. If at any time the patient presented recurrence of incontinence, at some point during the entire follow-up time of the patients, she was excluded from this study. This occurred in 109 patients out of 1000 who underwent surgery.

The patients were studied in the Salamanca University Assistance Complex and the Santísima Trinidad General Hospital in Salamanca (Spain). Total urinary continence, as measured subjectively and objectively by the ICIQ-SF test [18] and pad-test [19], respectively, was considered to mean success in urinary continence in the post-intervention controls at months 1, 6, and 12, and annually.

### 2.3. Study Groups

GA (*n* = 443): patients with anaesthetic risk ASA I.GB (*n* = 306): patients with anaesthetic risk ASA II.GC (*n* = 121): patients with anaesthetic risk ASA III.

### 2.4. Procedure

Patients were diagnosed with SUI by detailed anamnesis, physical examination, stress test, baseline blood and urine analysis, urine culture, urological ultrasound with post-voiding residue, validated ICIQ-SF questionnaire [18], health-related quality of life questionnaire SF-36 [20], the visual analog scale (EVA) [21], and the pad test [1]. Prior to surgery, an outpatient evaluation was performed by the Anaesthesiology Service, who used the ASA-PS Classification System [13] to estimate the risk posed by anaesthesia. Patients with previous urinary incontinence surgery, associated neurological pathologies, post-void residuals greater than 150 mL, or a history of pelvic radiotherapy were excluded.

The transobturator suburethral tape was placed by means of an anterior colpotomy 1 cm below the urinary meatus, followed by bilateral paraurethral dissection to the lower edge of the ischiopubic branch, in addition to a 5 mm incision at the level of the inguino-crural fold on both sides, at the level of the clitoris. A Kim System^®^ 1.5 × 30 cm Polypropylene mesh (Neomedic ©, 08225 Terrassa, Barcelona. Spain) was used. The mesh was installed at the level of the middle urethra, free of tension, with two helical needles inserted from the incisions of the inguinocrural folds to the anterior colpotomy. Regional anesthaesia was used for most cases and general anesthesia in the remainder. In all patients, antibiotic treatment was started 1 h pre-intervention and was continued until its completion 3 days after surgery. The prescribed analgesia was also maintained for 72 h. Ninety-seven percent of the cases were performed under a regimen of major ambulatory surgery (MAS) and required less than 24 h of hospital admission after the intervention.

In follow-up exams, a urological examination was performed to rule out mesh complications at the suburethral level and the presence of SUI through the stress test [22]. Patients were given ICIQ-SF [18], quality of life SF-36 [20], and visual analog scale (VAS) questionnaires [13] in post-intervention controls at months 1, 6, and 12, and annually. Scale of pain stimation was Visual Analog Scale (VAS) questionnaire. It was identical in all centers.

The pad-test was not carried out in women who did not present any signs or symptoms of incontinence according to the ICIQ-SF test during follow-up. Pad-tests were only performed on all patients preoperatively.

### 2.5. Variables Investigated

The following were assessed: age, body mass index, follow-up time, concomitant diseases, surgical history, obstetric-gynecological history, and toxic habits.

The complications derived from surgery that were found to influence functional recovery were:(a)local bleeding requiring vaginal tamponade or surgical revision of the site,(b)local postoperative pain requiring scheduled and rescue analgesia beyond the fourth day, and(c)infection of the surgical area needing antibiotic treatment beyond the fifth day.

Functional recovery after surgery was defined as physical recovery and the return to the patient’s normal pre-admission activity, with normal voiding without incontinence, continuing in a similar fashion for the first year, with the best possible conditions in daily living. Functional recovery in women of working age is found in the return to work. In women who do not have paid work, the recovery to function is to carry out the activities of daily life without limitations due to complications caused by the urinary incontinence surgery.

### 2.6. Statistical Analysis

The statistical package NCSS277/GESS2007 was used. Descriptive statistics, Student’s T, Chi2, Fisher, ANOVA, and multivariate analysis were applied. The Kolmogorov-Smirnov and Shapiro-Wilk tests were used to determine whether the distribution of the continuous variables was normal. Hypotheses were tested with non-parametric Chi-square tests and Kruskal-Wallis. The positive results of the latter were subanalyzed with Mann-Whitney U. A value of *p* < 0.05 was considered statistically significant.

### 2.7. Ethics

The study with code E.O.12/274 was approved by the Clinical Research Ethics Committee of the University Hospital of Salamanca.

## 3. Results

Mean age was 60.10 years, SD 13.38, median 60 (range 30–81), with no difference between the groups (*p* = 0.4852). In all groups the range was wide.

Mean body mass index (BMI) was 26.55kg/m2, SD 4.51, median 25.80, and range 17.96–45.78. BMI was lower in GA (mean 25.03, SD 3.23, median 25.39, range 17.96–27.56) than in GB (mean BMI 27.05, SD 5.01, median 25.80, range 17.96–35.67) and GC (mean 27.03, SD 3.86, median 26.80, range 21.21–45.78) (*p* = 0.0018).

Mean follow-up time was 2636.03 days, 831.35 SD, median 2555, and range 1460–5475. In GA the follow-up time was shorter (mean 24690.27 days, SD 803.03, median 2190 and range 1460–5475) than in GB (mean 2723.31 days, SD 824.60, median 2555 and range 1468–5475) and GC (mean 2904.57, 850.60 SD, median 2555 and range 1825–5110) (*p* = 0.0000001).

Group GB had more HTA (38.6%) than GC (23.23%) (*p* = 0.0012). There was more type 2 diabetes in GB (19.83%) than in GC (10.56%) (*p* = 0.013). Group GA had no cardiocirculatory pathology. There were more digestive disorders (gastritis and ulcus) in GB (29.75%) and GC (32.39%) than in GA (*p* = 0.0089). GB had more respiratory disorders (6.97%) than GC (2.11%) (*p* = 0.0325). There were more women with anxiety in GB (19.3%) than in GC (6.33%) (*p* = 0.0221) and GA (10.51%) (*p* = 0.0004). There was more hypothyroidism in GB (16.08%) than in GC (2.11%) (*p* = 0.0001) or GA (9.07%) (*p* = 0.0021). There was more history of curettage in GC (11.97%) versus GB (5.63%) (*p* = 0.0221), and more pelvic surgery in GB (71.31%) and GC (66.9%) compared to GA (32.57%) (*p* = 0.0001). There were more concomitant treatments with benzodiazepines in GC (27.46%) and GB (28.41%) than in GA (8.86%) (*p* = 0.02), and more parapharmacy products used in GB (17.96%) than in GC (6.33%) (*p* = 0.0007). Smoking and ex-smoking patients were 5.1% and 3.9%, respectively, with no differences between the groups (*p* = 0.2376). The absence of active smokers in GA is of note (Figure 1).

After the operation, 48 patients in GA (10.83%) had some sign or symptom requiring medical attention, 49 in GB (16.06%) and 16 in GC (13.22%). These complications (bleeding, pain, and infection) negatively affected functional recovery. In GA: bleeding: *n* = 3, pain *n* = 5, and infection *n* = 3; In GB: bleeding: *n* = 1, pain *n* = 13, and infection *n* = 12; In GC: bleeding: *n* = 3, pain *n* = 3, and infection *n* = 3 (Figure 2).

GC had more bleeding (18.75%) than GB (2.04%) (*p* = 0.0432). There was pain in 26.53% of GB and in 10.41% of GA, but the difference was not significant. (*p* = 0.0658). GB showed more infection (24.48%) than GA (6.25%) (*p* = 0.0224).

Table 1 shows the distribution of time to functional recovery in women who had bleeding, pain, or infection.

Mean time to functional recovery in GA women who had complications (bleeding, pain, or infection) was 5.72 days, standard deviation (SD) 2.05, median 7, and range 2–8.

In GA, the mean time to functional recovery in women with bleeding was 3 days, Standard Deviation (SD) 1, median 3, and range 2–4; in women with pain 6.40 days, SD 1.34, median 7, and range 5–8; and in women with infection 7.33 days, SD 0.57, median 7, and range 7–8. Patients who presented bleeding recovered function more quickly than those with pain or infection (*p* = 0.002944).

The mean functional recovery time in GB women who had complications (bleeding, pain, or infection) was 27.96 days, SD 28.42, median of 14, and a range of 3 to 90.

In GB, the mean time to functional recovery in women with bleeding was 3 days, SD0, median 3, range 3–3; in women with pain 46.69 days, SD 31.36, median 30, and range 5–90; in women with infection 10.83 days, SD 3.90, median 10, and range 7–20. Patients with bleeding recovered sooner (*p* = 0.002437). Dispersion was high due to the difference in recovery time between bleeding and pain, and is the complication with the greatest impact on functional recovery before patients can return to working life (mean: 46.69 days).

Mean functional recovery time in GC women who had complications (bleeding, pain, or infection) was 9.44 days, SD 2.50, median 10, and range 6–14.

In GC, the mean time to functional recovery in women with bleeding was 7.66 days, SD 2.08, median 7, and range 6–10; in women with pain 10.66 days, SD 1.15; median 10, and range 10–12; in women with infection 10 days, SD 3.46, median 8, and range 8–14. There were no differences between the different factors that delay functional recovery (*p* = 0.3482).

Mean time to functional recovery in patients with bleeding in GA, GB and GC was 4.16 days, SD of 1.94, median of 3.5, and range of 2 to 10; it was lower in GA (mean 3 days) and GB (mean 3 days) than in GC (mean 7.66 days) (*p* = 0.049) (Table 2).

Mean time to functional recovery in patients with pain in GA, GB and GC was 31.33 days, SD 30.70, median 14, and range 5–90; it is the factor that most affected the delay in functional recovery. The women in GB took the most time to return to work (45.96 days) compared to GA (6.4 days) and GC (10.66 days). (*p* = 0.01468) (Table 3).

Mean time to functional recovery in patients with infections in GA, GB and GC was 10.11 days, SD 3.61 days, median 9, and range 7–20. The mean time to functional recovery in GA was 7.33 days, SD 0.47, median 7, and range 7–8; in GB 10.27 days, SD 3.57, median 9, and range 7–20; and in GC 10 days, SD 2.44, median 9, and range 8–14. There were no differences between the groups (*p* = 0.343577).

General anesthesia was used in 20 patients from the entire sample. Spinal anesthesia was used in the rest of the patients. No tape was implanted under local anesthesia. No relationship was found between the type of anesthesia used and the appearance of procedural complications.

## 4. Discussion

### 4.1. Functional Recovery and Age

Our results agree with other studies that indicate that age is not a risk factor for complications in surgery. Frailty, which refers to biological rather than chronological age, is more important in association with postoperative complications. Other studies on patients operated on in Major Ambulatory Surgery (MAS) agree with our results [23,24]. Therefore, we conclude that age is not a risk factor in the ASA-PS classification.

The main reason that patients who had surgery failure, in any degree of persistent urinary incontinence, was that they were older women, multiple comorbidities (many more than women who were successful with surgery), and in 25 cases (22.93%) they had been surgically treated for their incontinence. This posed a real bias problem. Therefore, the authors decided to exclude these patients. However, this group of 109 patients will be the object of another study to find out what factors could be related to both the failure of surgery and the functional recovery time of the patients.

### 4.2. Functional Recovery and BMI

BMI was lower in ASA I. Obesity is associated with an increased risk of postoperative complications, unplanned admissions, and cancellations in ambulatory surgery [24,25]. Proper patient selection, preoperative evaluation, and prevention and control strategies for the most frequent complications in obese patients are key factors. Patients must be appropriately selected according to BMI, not exceeding 30 in ASA I or 40 in ASA II, as studies show this is an important risk factor for postoperative and MAS complications, although these studies include some patients with BMI over 40 in ASA II or over 30 in ASA I [25].

### 4.3. Functional Recovery and Complications

This is a study with a long follow-up time. Although the follow-up period was shorter in GA, this has no clinical impact or bias, since the range is between 4 and 15 years, much higher than other follow-up studies of women operated on for SUI [26].

The study presented is an analysis of the functional recovery time of a sample of women who have received surgical treatment for urinary incontinence using TOT. We wanted to have a minimum follow-up time in all patients of 1 year to rule out complications related to the device. It is a common procedure of this research group that after implanting a pelvic floor biomaterial, wait at least one year to observe or rule out complications related not to the surgical procedure, but to the device itself. The authors have found that there were no late complications attributable to the device (Kim System^®^ suburethral tape, Neomedic International ©) that were related to the postoperative functional recovery of the patients. The implanted devices did not cause complications over time.

The risk of postoperative infection has been reported to increase with age, with conditions such as diabetes, or with a higher ASA risk [15]. In our study, infection was more prevalent in women at risk of ASA II, and more frequent in those with allergies to antibiotics, with the result that antibiotic coverage was conditioned by allergies and deviated from protocol. We describe the association between type of surgery, ASA level and the presence of infection in the postoperative period, describing a rate of infection in the surgical suite and validating the risk stratification of the National Nosocomial Infection Surveillance Index (NNIS) and the influence of each factor. The duration of the intervention, specifically, is considered the factor that most influences infection rates, and ASA level the least [27]. In our study, the duration of the procedure was similar in all women, so this factor cannot be evaluated.

Other studies indicate that the most frequent complications after TOT tape surgery are infection and pain [28]. Postoperative acute pain is a subjective, multidimensional experience, is complex to address, and one that requires a global and multidisciplinary approach. The American Association of Anesthesiology defines it as pain that is present in a surgical patient as a result of the disease, the surgical procedure and its complications, or a combination of both [29]. Pain remains under-treated on occasion despite advances in our understanding of nociception, new drugs, and new analgesic techniques. Its prevalence varies, being, in our study and among the women who presented some postoperative sign or symptom for which they consulted, 10.41% in ASA I, 26.53% in ASA II and 18.75% in ASA III, with no difference between the groups (*p* = 0.0658). It is difficult to estimate the overall prevalence of acute postoperative pain in major outpatient surgery, as many types of procedures are performed; some studies estimate a prevalence of up to 30% of patients with moderate-severe pain [29]. Our procedure is rated, according to the intensity of pain expected, as mild, with a visual analog scale (VAS) of up to 4 points in some studies [30]. The implementation of appropriate protocols for each procedure is important for improving postoperative pain [31].

### 4.4. Time to Functional Recovery

The time to functional recovery in cases of no complications delaying the return to routine life was 5.10 days, with no differences found between the groups.

In patients with complications, mean time to recovery was 16.91 days, and was conditioned by factors such as bleeding, pain, and infection. In GA, the mean was 5.72 days, and range 2–8. Bleeding was the complication that least affected their return to work, a maximum of 4 days after it occurred. In GB, the mean time to recovery was 27.96 days, and range 2–90. In GB, times were highly dispersed, conditioned by pain, with a mean of 46.88 days, compared to bleeding patients who had a mean of 3 days. In the GC group, the mean was 9.66 (range 6–14) days until functional recovery, with very little dispersion and no difference between the different factors that delay functional recovery.

When comparing among the ASA groups according to each complication that conditions functional recovery, in bleeding we found a mean of 4.1 (range 2–10) days; the dispersion is very low, but nevertheless there is a difference in functional recovery, being earlier in GA than in GC, which has a mean of 7.66 days (range 6–10). These findings are in agreement with the work published by Voney et al. [15], where the higher the ASA, the greater the risk of postoperative complications, including bleeding and the state of frailty, a characteristic which is more present in ASA III patients.

Pain is the factor that most determines the delay in functional recovery. The mean time was 29.16 days, with a great dispersion among the patients (range 5–90). It was found that in GB, patients took longer to return to normal activity. Pain aggravates functional impairment; it limits daily activities, mobilization, and the ability to participate in postoperative rehabilitation, delaying return to working life; and it may contribute to the development of chronic pain [32]. Other studies agree with our results, and list conditions such as depression, anxiety and psychological vulnerability as predisposing factors for postoperative pain [33]; there are also those who argue that several comorbidities such as obesity, dyslipidemia and high blood pressure, in which the organism is in a permanent systemic inflammatory process, or stresses such as trauma and previous surgeries, can trigger more pronounced pain [34,35].

In infection cases, recovery took a mean of 9.60 days, range 7–20, with no differences between groups. Although infection is a greatly feared complication, it has obvious symptoms and signs, so treatment and management are very similar in all patients and dispersion is therefore very low. ASA risk is one of the items that least influences the number of infections, according to several studies [27]. There are similar studies where the incidence of infection is also not high, although certain factors are associated with an increased risk of infection. Vigil et al. studied risk factors for urinary tract infection after surgery with a suburethral sling. Factors independently associated with an increased risk of infection included: age greater than 65 years, BMI greater than 40 kg/m2, and hospital admission [36]. We have not corroborated this association in our series.

The evaluation of recovery time has been measured in a standardized way in all patients.

It has been a very important finding in the study to find that GB patients, who have ASA II risk and in theory have a better general state of health than those of the CG group, who have ASA III risk, are more likely to present delay in the return to normal life in relation to presenting disabling pain levels more frequently. The authors hypothesize that CG women probably have better tolerance to postoperative pain.

On the other hand, it is pointed out that depression, anxiety and hypertension are associated with a slower recovery. These three diagnoses are more frequent in GB than in CG.

Therefore, this is a main finding of the study.

Regarding the administration of 7 days of antibiotics after surgery, in the participating centers, in recent years an attempt has been made to change the standardized system of completing 1 week of antibiotic coverage after the placement of a biomaterial, especially when the approach (transvaginal) is contaminated.

Therefore, this is a “classic” system of hospitals where surgical interventions have been performed, which is expected to change in the immediate future.

### 4.5. Proposal for Improvement

The proposal has two stages:

STAGE A:

Evaluate variables related to the risk of pain mismanagement, intolerance and/or persistence of pain, beyond the ASA-PS cataloguing.

Special attention to:High blood pressure.Diabetes mellitus type 2.Hypothyroidism.Psychological disorders (anxiety, depression).Respiratory disorders.Regular use of parapharmacy.Obesity.History of pelvic surgery.

STAGE B:

Multidisciplinary treatment of pain and anxiety beginning preoperatively, with anxiolytics and analgesics if necessary, continuing in the intraoperative phase with anaesthetic techniques, and ending in the postoperative period with the prescription of a standardised and protocolised treatment. Follow up by telephone to verify that the patient respects the treatment, with protocols that attempt to keep the VAS always below 3. Avoid indications of the type “if pain”, “if needed”; instead, medication must be strictly prescribed.

Monitor the patient more closely.

The study has led the authors to the need to improve the management of women treated surgically for stress urinary incontinence using TOT to shorten recovery time. How? Implementing these two stages of better control of possible complications related to the procedure that lead to a delay in functional recovery. The implementation and monitoring of the results of the improvement of the protocol will be evaluated in a subsequent study where the results of the “conventional care” stage are compared with the results of the “improved care, taking into account the objective of early functional recovery”. 

## 5. Conclusions

For female patients with stress urinary incontinence treated with transobturator suburethral tape who achieve total urinary continence without complications, the average time to return to normal activity is 5 days. In patients who have complications related to the surgical procedure, the average recovery time is 16.91 days and is conditioned by factors such as bleeding, pain, and infection.

Pain is the most frequent complication, and ASA II patients take longer to recover than ASA I and ASA III patients. Pain is the complication that most conditions the delay in functional recovery. Bleeding is the least frequent, and ASA I and II patients recover earlier than ASA III. In the complication of infection, there is no difference in recovery time relative to ASA risk.

Classification into ASA-SP risk groups can be used to anticipate functional results in women undergoing surgery for urinary incontinence with transobturator suburethral tape. An ASA-PS risk-based functional recovery forecasting protocol should be adopted, especially to identify which ASA II patients may have prolonged invalidating postoperative pain.

We propose preventive management measures that favour functional recovery.

## Figures and Tables

**Figure 1 jcm-09-02607-f001:**
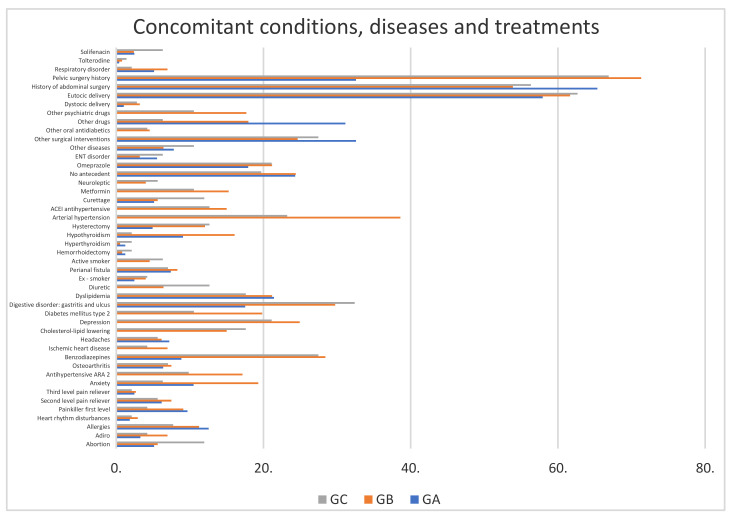
Concomitant conditions, diseases and treatments in GA, GB and GC (% of affected women). GA: patients with anaesthetic risk ASA I. GB: patients with anaesthetic risk ASA II. GC: patients with anaesthetic risk ASA III.

**Figure 2 jcm-09-02607-f002:**
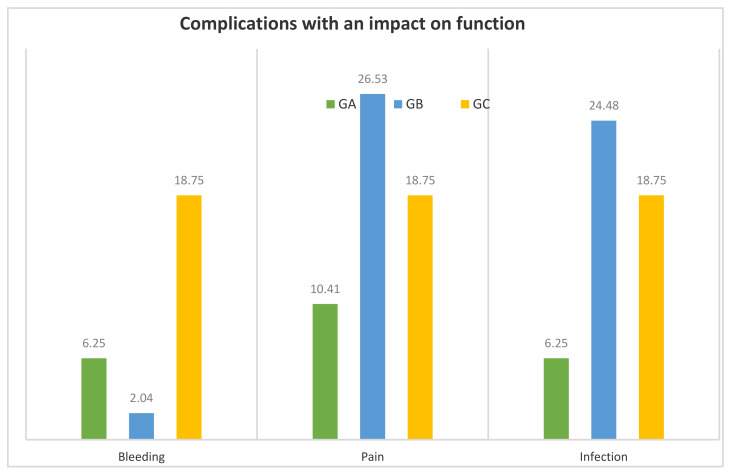
Complications with an impact on function (% of women who showed some sign or symptom requiring medical attention) in GA, GB and GC. GA: patients with anaesthetic risk ASA I. GB: patients with anaesthetic risk ASA II. GC: patients with anaesthetic risk ASA III.

**Table 1 jcm-09-02607-t001:** Days until functional recovery by complication (*n*).

Days until Functional Recovery	Bleeding	Pain	Infection
GA	2	5	7
	3	5	7
	4	7	8
		7	
		8	
GB	3	5	7
		5	7
		14	7
		30	7
		30	10
		30	10
		30	10
		60	12
		60	12
		60	14
		90	14
		90	20
		90	
GC	6	10	8
	7	10	8
	10	12	14

*n*: number of days until functional recovery in each of the patients in whom the complication occurred. GA: patients with anaesthetic risk ASA I. GB: patients with anaesthetic risk ASA II. GC: patients with anaesthetic risk ASA III.

**Table 2 jcm-09-02607-t002:** Time to functional recovery (days) in patients with bleeding.

Bleeding which Delays Functional Recovery	Mean Time in Days	Standard Deviation (SD)	Median	Range
**GA**	3	1	3	2–4
**GB**	3	−	3	−
**GC**	7.66	2.08	7	6–10

GA: patients with anaesthetic risk ASA I. GB: patients with anaesthetic risk ASA II. GC: patients with anaesthetic risk ASA III.

**Table 3 jcm-09-02607-t003:** Time to functional recovery (days) in patients with pain.

Pain which Delays Functional Recovery	Mean Time in Days	Standard Deviation (SD)	Median	Range
**GA**	6.40	1.34	7	5–8
**GB**	45.69	31.36	30	5–90
**GC**	10.66	1.15	10	10–12

GA: patients with anaesthetic risk ASA I. GB: patients with anaesthetic risk ASA II. GC: patients with anaesthetic risk ASA III.

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
