# Peer review of "The American Society of Anesthesiologists Physical Status (ASA-PS) Risk Group Classification Can Be Used to Anticipate Functional Recovery Outcomes after the Surgical Treatment of Female Urinary Incontinence with Transobturator Suburethral Tape"

_jcm, 2020, doi:10.3390/jcm9082607_

Round 1

Reviewer 1 Report

the paper concerns very important topic as the mid-urethral sling seems to be a gold standard of sui treatment. 

although there are some points to be developed or explain.

  1. what the authors mean by functional recovery - it should be revealed
  2. what is the reason according to authors of such high recovery time in GB group - inadequate to other groups? it may be a big bias of the study.
  3. what is the reason for continuation antibiotics for 3 days after the surgery. Is it local recommendation?
  4. what was the scale of pain estimation? was it identical in all centres?
  5. were there a differences in complication between the type of the anaesthesia? was the pain worse in case of general or local an?

Author Response

Good morning

Thank you for your comments and appreciations. Below we give an answer to the questions raised:

1.-what the authors mean by functional recovery - it should be revealed

ANSWER:

Functional recovery: the concept or meaning of functional recovery is expressed in the abstract:

Introduction: …functional recovery, undertood as the return to routine life prior to the intervention”.

It is defined in paragraph 7 of the Introduction:

Functional recovery is defined as the physical recovery and functional progression of all aspects affected by an injury, and is a fundamental element for returning to work or normal activity under the best possible conditions, safely and painlessly [17].

It is defined in the investigated variables (point 2.5.):

Functional recovery after surgery was defined as physical recovery and the return to the patient’s normal pre-admission activity, with normal voiding without incontinence, continuing in a similar fashion for the first year, with the best possible conditions in daily living.

According to the observation of reviewer 1, it is added in section 2.5. (page 4) this paragraph: Functional recovery in women of working age is found in the return to work. In women who do not have paid work, the recovery to function is to carry out the activities of daily life without limitations due to complications caused by the urinary incontinence surgery.

2.-what is the reason according to authors of such high recovery time in GB group - inadequate to other groups? it may be a big bias of the study.

ANSWER:

IT IS ADDED IN THE DISCUSSION SECTION, SECTION “ Time to functional recovery”,  after PARAGRAPH 4 (page 10) this paragraph:The evaluation of recovery time has been measured in a standardized way in all patients.It has been a very important finding in the study to find that GB patients, who have ASA II risk and in theory have a better general state of health than those of the CG group, who have ASA III risk, are more likely to present delay in the return to normal life in relation to presenting disabling pain levels more frequently. The authors hypothesize that CG women probably have better tolerance to postoperative pain.On the other hand, it is pointed out that depression, anxiety and hypertension are associated with a slower recovery. These 3 diagnoses are more frequent in GB than in CG.Therefore, this is a main finding of the study. 

3.-what is the reason for continuation antibiotics for 3 days after the surgery. Is it local recommendation?

ANSWER: This paragraph is added at the end of the discussion:Regarding the administration of 7 days of antibiotics after surgery, in the participating centers, in recent years an attempt has been made to change the standardized system of completing one week of antibiotic coverage after the placement of a biomaterial, especially when the approach (transvaginal) is contaminated.Therefore, this is a "classic" system of hospitals where surgical interventions have been performed, which is expected to change in the immediate future.

4.-what was the scale of pain estimation? was it identical in all centres?

En el apartado de Material  y métodos, punto 2.4.Procedure (page 4), this paragraph is added:

Scale of pain stimation was Visual Analog Scale (VAS) questionnaire. It was identical  in all centers.

5.-were there a differences in complication between the type of the anaesthesia? was the pain worse in case of general or local an?

At the end of the Results section (page 8), this paragraph is added:

General anesthesia was used in 20 patients from the entire sample. Spinal anesthesia was used in the rest of the patients. No tape was implanted under local anesthesia. No relationship was found between the type of anesthesia used and the appearance of procedural complications.

Reviewer 2 Report

This is a non-concurrent prospective observational multicenter study to assess time to functional recovery after TOT surgery for SUI who had complete continence after surgery.

  1. Under Sampling:

Total urinary continence, as measured subjectively and objectively by the ICIQ-SF test [18] and pad-test [19] respectively, was considered to mean success in urinary continence in the post-intervention controls at months 1, 6, and 12, and annually

  1. Under Procedure

the presence of SUI through the stress test [22]. Patients were given ICIQ-SF [18], quality of life SF-36 [20], and visual analog scale (VAS) questionnaires [13] in post-intervention controls at months 1, 6, and 12, and annually.

  • Second statement Under Procedure does not include pad-test, is this just oversight?
  • At which point on follow-up was the total continence defined? 1, 6, 12 or more months? What about patient who were dry at 6 months but wet at 12 months?
  • What was the reason to collect information after 1 or more years when objective of the study was to calculate days to functional recovery and the longest time was 90 days.
  • It would be of interest to see data of 109 patients who were not completely dry. Some degree of post-surgery incontinence is acceptable to the patient
  • The title “Management protocol for the surgical treatment of female urinary incontinence to shorten the time of functional recovery” does not reflect the objectives and conclusions. Conclusions are “The ASA-SP risk group classification can be used to anticipate functional outcomes” There is no data given to the 2-stage proposal how to shorten the time to functional recovery as the title indicate. This would require new study to show that the propose 2-stage measures has an effect on the time to functional recovery.

Author Response

1.-Under Sampling:

Total urinary continence, as measured subjectively and objectively by the ICIQ-SF test [18] and pad-test [19] respectively, was considered to mean success in urinary continence in the post-intervention controls at months 1, 6, and 12, and annually

2.-Under Procedure

the presence of SUI through the stress test [22]. Patients were given ICIQ-SF [18], quality of life SF-36 [20], and visual analog scale (VAS) questionnaires [13] in post-intervention controls at months 1, 6, and 12, and annually.

1.-Second statement Under Procedure does not include pad-test, is this just oversight?

Answer: in section 2.4.-Procedure:

It is added at the end of the section:

The pad-test was not carried out in women who did not present any signs or symptoms of incontinence according to the ICIQ-SF test during folow-up. Pad-tests were only performed on all patients preoperatively.

2-At which point on follow-up was the total continence defined? 1, 6, 12 or more months? What about patient who were dry at 6 months but wet at 12 months?

In section 2.2. of Material and methods, this paragraph is added at the end of the first paragraph (page 3):Continence was evaluated postoperatively at month 1, 6, 12 and annually. If at any time the patient presented recurrence of incontinence, at some point during the entire follow-up time of the patients, she was excluded from this study. This occurred in 109 patients out of 1000 who underwent surgery.

3.-What was the reason to collect information after 1 or more years when objective of the study was to calculate days to functional recovery and the longest time was 90 days.

In section 4.-Dicussion, in section Functional recovery and complications
, is added after paragraph 1 (page 9): •

The study presented is an analysis of the functional recovery time of a sample of women who have received surgical treatment for urinary incontinence using tot. We wanted to have a minimum follow-up time in all patients of one year to rule out complications related to the device. It is a common procedure of this research group that after implanting a pelvic floor biomaterial, wait at least one year to observe or rule out complications related not to the surgical procedure, but to the device itself. The authors have found that there were no late complications attributable to the device (Kim System® suburethral tape, Neomedic International ©) that were related to the postoperative functional recovery of the patients. The implanted devices did not cause complications over time.

4.-It would be of interest to see data of 109 patients who were not completely dry. Some degree of post-surgery incontinence is acceptable to the patient.

In section 4.-Dicussion, section Functional recovery and age, is added after paragraph 1 (page 8):• The main reason that patients who had surgery failure, in any degree of persistent urinary incontinence, was that they were older women, multiple comorbidities (many more than women who were successful with surgery), and in 25 cases (22.93%) they had been surgically treated for their incontinence. This posed a real bias problem. Therefore the authors decided to exclude these patients. However, this group of 109 patients will be the object of another study to find out what factors could be related to both the failure of surgery and the functional recovery time of the patients.

5.-The title “Management protocol for the surgical treatment of female urinary incontinence to shorten the time of functional recovery” does not reflect the objectives and conclusions. Conclusions are “The ASA-SP risk group classification can be used to anticipate functional outcomes”

The reviewer is correct, therefore we change the title to:

  • “TheAmerican Society of Anesthesiologists Physical Status (ASA-PS)risk group classification can be used to anticipate functional recovery outcomes in the surgical treatment of female urinary incontinence with transobturator suburethral tape”.

6.-There is no data given to the 2-stage proposal how to shorten the time to functional recovery as the title indicate. This would require new study to show that the propose 2-stage measures has an effect on the time to functional recovery.

At the end of the section  5.-Proposal for improvement:
the authors add this section: • The study has led the authors to the need to improve the management of women treated surgically for stress urinary incontinence using TOT to shorten recovery time. How? Implementing these two stages of better control of possible complications related to the procedure that lead to a delay in functional recovery. • The implementation and monitoring of the results of the improvement of the protocol will be evaluated in a subsequent study where the results of the “conventional care” stage are compared with the results of the “improved care, taking into account the objective of early functional recovery”.